# Risk of preterm birth associated with maternal gingival inflammation and oral hygiene behaviours in rural Nepal: a community-based, prospective cohort study

Daniel J Erchick [ID],[1] Subarna K Khatry,[2] Nitin K Agrawal,[3] Joanne Katz,[1] Steven C LeClerq,[1,2] Bhola Rai,[2] Mark A Reynolds,[4] Luke C Mullany[1]

¹Department of International Health, Johns Hopkins Bloomberg School of Public Health, Baltimore, Maryland, USA
²Nepal Nutrition Intervention Project - Sarlahi, Kathmandu, Nepal
³Department of Dentistry, Institute of Medicine, Tribhuvan University, Kathmandu, Nepal
⁴Department of Advanced Oral Sciences and Therapeutics, University of Maryland School of Dentistry, Baltimore, Maryland, USA

**Correspondence to**
Dr Daniel J Erchick;
derchick@jhu.edu

## ABSTRACT

**Objectives** Observational studies have identified associations between periodontitis and adverse pregnancy outcomes, but randomised controlled trials evaluating the efficacy of periodontal therapy have yielded inconsistent results. Few studies have explored relationships between gingival inflammation and these outcomes or been conducted in rural, low-income communities, where confounding risk factors differ from other settings.

**Methods** We conducted a community-based, prospective cohort study with the aim of estimating associations between the extent of gingival inflammation in pregnant women and incidence of preterm birth in rural Nepal. Our primary exposure was gingival inflammation, defined as bleeding on probing (BOP) ≥10%, stratified by BOP <30% and BOP ≥30%. A secondary exposure, mild periodontitis, was defined as ≥2 interproximal sites with probing depth (PD) ≥4 mm (different teeth) or one site with PD ≥5 mm. Our primary outcome was preterm birth (<37 weeks gestation). We used Poisson regression to model this relationship, adjusting for potential confounders.

**Results** Of 1394 participants, 554 (39.7%) had gingival inflammation, 54 (3.9%) mild periodontitis and 197 (14.1%) delivered preterm. In the adjusted regression model, increasing extent of gingival inflammation was associated with a non-significant increase in risk of preterm birth (BOP ≥30% vs no BOP: adjusted relative risk (aRR) 1.37, 95% CI: 0.81 to 2.32). A secondary analysis, stratifying participants by when in pregnancy their oral health status was assessed, showed an association between gingival inflammation and preterm birth among women examined in their first trimester (BOP ≥30% vs no BOP: aRR 2.57, 95% CI: 1.11 to 5.95), but not later in pregnancy (BOP ≥30% vs no BOP: aRR 1.05, 95% CI: 0.52 to 2.11).

**Conclusions** Gingival inflammation in women examined early in pregnancy and poor oral hygiene behaviours were risk factors for preterm birth. Future studies should evaluate community-based oral health interventions that specifically target gingival inflammation, delivered early in or before pregnancy, on preterm birth.

**Trial registration number** Nepal Oil Massage Study, NCT01177111.

### Strengths and limitations of this study

► This study collected data through a large, community-based, prospective cohort in a rural, low-resource setting.
► The study population had low prevalence of some established confounders of the relationship of interest, which are common in other populations.
► Some clinical periodontal measures were not collected due to visit time constraints, for example, recession on interproximal sites or gingival index score.
► Preterm birth classification was based on maternal self-report of last menstrual period instead of the gold standard, ultrasound examination.

## INTRODUCTION

Annually, 2.5 million babies die prior to 28 days of life, and preterm birth is the leading cause of these deaths.[1] Preterm newborns that survive are at substantial risk of mortality from other causes, long-term disabilities such as neurological and developmental impairments, and non-communicable diseases.[2] In low- and middle-income countries (LMICs), where the majority of preterm births occur, therapeutic interventions are often unavailable and difficult to scale up, especially in communities where many mothers deliver at home or in primary facilities without skilled care (eg, South Asia).[3]

Periodontal disease includes several inflammatory conditions, typically initiated by oral bacteria, beginning with reversible accumulation of plaque and inflammation of gingival tissue (gingivitis) and progressing to irreversible breakdown of the supportive tissues of the teeth and tooth loss (periodontitis).[4] Onset of new and worsening of existing gingival inflammation during pregnancy are normal

and well documented, peaking in the second or third trimester.[5] [6] Major physiological and hormonal changes occur during pregnancy with wide ranging effects on the body, including increased permeability of gingival capillaries, altered immune system activity and shifts in composition of the sub-gingival microbiome, including proliferation of aggressive bacteria associated with periodontitis, such as *Porphyromonas gingivalis*.[7] [8]

Periodontal disease in pregnant women has been associated with preterm birth and other adverse pregnancy outcomes.[9–11] Yet randomised controlled trials (RCTs) evaluating the impact of periodontal therapy during pregnancy on adverse pregnancy outcomes have produced inconsistent results.[9] [12–18] One meta-analysis found a significant effect of periodontal therapy among women at high risk of preterm birth.[19] Although the mechanisms underlying this observed association are unclear, hypotheses include haematogenic translocation of periodontal pathogens or their byproducts to the fetal-placental unit, or action of inflammatory mediators in the periodontium on levels of systemic inflammation.[20]

Alternatively, the observed relationship between periodontal disease in pregnant women and adverse pregnancy outcomes could be the result of unmeasured and uncontrolled confounding factors. Previously described confounders of this relationship, which are commonly controlled for in studies, include age, smoking, multiple birth, previous adverse pregnancy outcomes, ethnicity and socioeconomic factors.[11] Some studies, however, have proposed the possibility that a genetic inflammatory phenotype could be responsible for increased risk of periodontal disease, or failed periodontal therapy, and adverse pregnancy outcomes, particularly spontaneous preterm birth.[21]

Few studies have evaluated whether gingival inflammation is associated with risk of preterm birth or other adverse pregnancy outcomes. Further, studies of this association have been nearly universally facility-based, whether in high-income or low-income settings. Understanding this relationship from a population-based perspective in low resource communities can offer certain benefits from an epidemiological perspective. Many populations in LMICs have lower prevalence of important confounding factors of this relationship, such as smoking, alcohol use and chronic diseases (eg, hypertension or diabetes). Community-based studies can avoid selection bias associated with hospital-based studies, particularly in populations where home delivery remains common, as is the case in much of South Asia. Given these potential benefits, we conducted a community-based, prospective cohort study to estimate the association between gingival inflammation and preterm birth among women in a rural community in the Terai (plains) region of Nepal.

## METHODS

We conducted a community-based, prospective cohort study of maternal gingival inflammation and adverse pregnancy outcomes across nine village development committees in Sarlahi District, Nepal. Eligible participants included married pregnant women between the ages of 15 and 40 who were <26 weeks gestation at the time of enrolment. Participants were identified and determined eligible between January and November 2016 using the infrastructure of a large community-based randomised trial of topical applications for newborn massage, the Nepal Oil Massage Study (NOMS) (NCT01177111), which was actively enrolling a population-based sample of pregnant women in this study area.

### Study visits

Study visits were conducted in participant homes because of the wide dispersion of households across this rural community and the impracticality of bringing participants to a central location. Data on participant demographics, vital signs and morbidities during pregnancy, oral hygiene practices, care-seeking and knowledge, and other characteristics were collected through a series of questionnaires administered over several visits during the course of pregnancy. Data collection teams were notified of the birth outcome by a locally resident study staff member, and the date of birth and other data concerning the mother and newborn were collected as soon as possible after delivery.

Oral health examinations were performed by five auxiliary nurse midwives who were trained for the purpose of this study. Their training lasted 3 to 4 weeks and included classroom instruction and practice of periodontal techniques conducted by an experienced dentist (NKA) at the Department of Dentistry, Institute of Medicine, Tribhuvan University, Kathmandu, Nepal. We estimated the validity of probing depth (PD) measurements of the auxiliary nurse midwives relative to the dentist (NKA), finding per cent agreement, weighted kappa scores and intraclass correlation coefficients, with an allowance of PD ±1 mm, exceeded 99%, 0.7 and 0.9, respectively.[22]

### Periodontal measurements

Auxiliary nurse midwives used portable dental equipment to conduct a full mouth examination. Periodontal measurements were made using a colour Williams probe (Hu-Friedy, Chicago, Illinois, USA). PD was measured on six sites per tooth (disto, mid and mesial aspects of buccal and lingual surfaces), and the cemento-enamel junction to gingival margin (CEJ-GM) distance was measured on two sites per tooth (mid-buccal and lingual aspects), excluding third molars. Presence or absence of bleeding on probing (BOP) was recorded for any buccal or lingual probing site of each tooth. PD values were recorded in millimetres rounded to the next higher whole number. CEJ-GM distances were recorded similarly. If the free gingiva was coronal to the CEJ, the CEJ-GM measurement was recorded as 0. Clinical attachment loss (CAL) was calculated by summing PD and CEJ-GM distances.

## Outcome/exposure definitions

Our primary exposure definition, gingival inflammation, was defined according to a classification scheme developed by the American Academy of Periodontology (AAP) and European Federation of Periodontology to update the 1999 classification of periodontal diseases and conditions.[23] Clinical health was defined as all sites PD ≤3 mm and BOP <10%.[24 25] Gingival inflammation was defined as BOP ≥10%, stratified as localised gingival inflammation (BOP 10% to 30%) and generalised gingival inflammation (BOP ≥30%).[24–29]

We defined secondary exposure definitions for mild and moderate periodontitis based on a modified version of Centers for Disease Control and AAP updated 2012 case definitions for population-based surveillance of periodontitis.[28 30] Mild periodontitis was defined as ≥2 interproximal sites with PD ≥4 mm (not on the same tooth) or one site with PD ≥5 mm. Moderate periodontitis was defined as ≥2 interproximal sites with PD ≥5 mm (not on the same tooth). CAL was not included in these definitions as we did not collect this measure at interproximal sites to shorten the visit time; however, clinical attachment level and probing depth are widely considered equivalent measures of periodontitis in younger adults.[31]

Gestational age was calculated using the last menstrual period (LMP) method as recalled by the mother at 5-weekly pregnancy surveillance home visits. Our 5-weekly pregnancy surveillance approach has several benefits, including short recall (≤5 weeks) and pregnancy testing, as compared with traditional LMP approaches, which often rely on LMP recall later in pregnancy or at time of delivery. Preterm birth was defined as a live birth or stillbirth <37 completed weeks of gestation at the time of delivery.

## Statistical analysis

Bivariate analyses between participant characteristics and the outcome, preterm birth, were evaluated using t-tests and Poisson regression for continuous and binary and categorical variables, respectively. We calculated unadjusted relative risks and adjusted relative risks (RR and aRR) of preterm birth and associated 95% CIs using Poisson regression with robust variance, which was used due to occurrence of convergence issues with other regression methods. Given our sample size (n=1394) and assuming a power of 80%, type I error of 0.05, and our pre-study estimates of population prevalence of preterm birth (17%) and gingival inflammation (40%), we estimated that we could detect a relative risk of preterm birth of roughly >1.5.

Multivariable models were constructed by sequentially adding groups of covariates, including maternal characteristics, oral hygiene behaviours and socioeconomic factors. Covariates associated with preterm birth at the p<0.10 level in bivariate analyses were considered in these regression models. Additional variables, known through previous studies to be confounders of the periodontal disease and preterm birth relationship, were also included in regression models (including age, ethnicity, body mass index, primiparity, multiple births and socioeconomic variables (ie, literacy, education and indicators of household wealth).

In an effort to remove a possibly attenuating effect of including cases of transient pregnancy-induced gingival inflammation, we examined the relationship stratified by timing of the oral exam. Women were categorised into two trimester groups (<13 weeks and ≥13 to <26 weeks gestation). Similar to the primary analysis, for each trimester group we calculated aRR of preterm birth and associated 95% CIs using Poisson regression with robust variance.

All statistical analyses were performed in Stata 14.2 (StataCorp, College Station, Texas, USA). All participants provided written consent for this study.

## Patient and public involvement

There was no patient or public involvement in this study.

## RESULTS

### Participant characteristics

Between 11 January 2016 and 26 November 2016, among 2821 pregnancies in the study area, 2291 (81.2%) participants were identified by the parent trial as eligible for enrolment (<26 weeks gestation) (figure 1). Among eligible participants, 1478 (64.5%) were enrolled and six refused participation (0.3%). Another 807 (35.2%) were not visited for the purposes of the study because the eligible participants in the study area initially exceeded the capacity of the auxiliary nurse midwives to enrol women before their gestational age exceeded the exclusion criterion of <26 weeks. For logistical reasons, enrolment of women earlier in pregnancy was prioritised at the start of study enrolment, while enrolment of women later in pregnancy was prioritised towards the end of study enrolment. However, among eligible women, the included and excluded participant groups did not differ by age, education or socioeconomic factors.

Birth outcomes for the 1474 women followed until the end of pregnancy were recorded as 1345 single live births, 14 twin live births, 33 single stillbirths, 1 twin stillbirth, 1 set of twins where 1 was live born and 1 stillborn, 74 miscarriages and 6 abortions. As our primary outcome was preterm birth, pregnancy was selected as the unit of analysis (and, therefore, twins were counted only once). There were 1394 total pregnancies for analysis, of which 197 (14.1%) were preterm. Preterm deliveries included 163 (11.7%) moderate preterm (<37 to ≥32 weeks), 27 (1.9%) very preterm (<32 to ≥28 weeks) and 7 (0.5%) extremely preterm (<28 weeks). Over three-quarters of participants (n=1053, 75.5%) were full-term (≥37 weeks) and 144 (10.3%) were post-term (≥42 weeks).

Baseline demographic, periodontal, oral hygiene and dental healthcare seeking characteristics of participants by extent of gingival inflammation were previously reported.[32] At enrolment, mean gestation was 14.5 (SD: 4.3) weeks (range: 6.4 to 27.7 weeks), with 574 women in

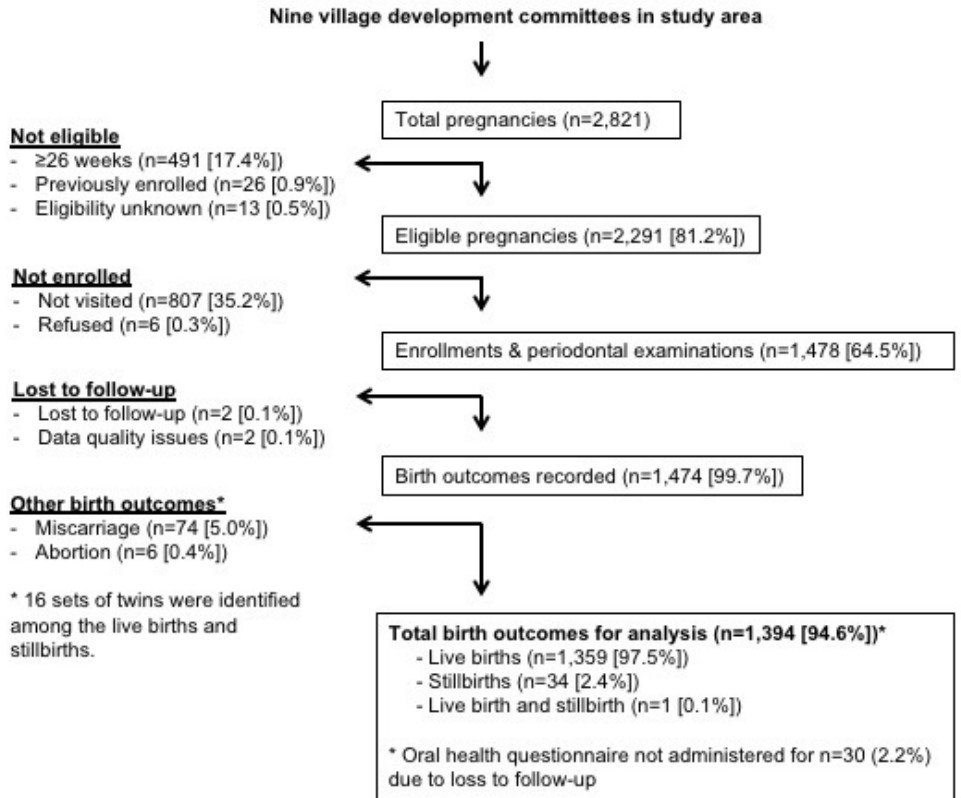

**Figure 1** Cohort study participation flow chart. Data on eligible pregnancies, enrolled pregnancies, refusals, lost to follow-up and birth outcomes.

their first trimester and 820 in their second. Participant age averaged 23.0±4.6 years, and risk of preterm birth was higher (RR 1.93, CI: 1.01 to 3.72) for women ≥35 years relative to those 18 to 35 years (table 1). Several known confounders of the gingival inflammation and preterm birth relationship had a reported prevalence near 0% in this study population, including smoking and other tobacco use, alcohol use and hypertension.

According to our definition, 554 (39.7%) participants had gingival inflammation and 840 (60.3%) were healthy (table 2). Of those with gingival inflammation, 445 (80.3%) had localised gingival inflammation (BOP <30%) and 109 (19.7%) generalised gingival inflammation (BOP ≥30%). Most participants (n=1105, 79.3%) had at least one site with BOP. According to our secondary exposure definitions, 54 participants (3.9%) had mild and 4 (0.3%) had moderate periodontitis. More participants (n=120, 8.6%) had ≥1 site with PD ≥4 mm. Mean CAL was equivalent to mean PD, with nearly all of CAL ≥4 mm due to pocketing secondary to gingival enlargement, indicating very little recession of the gingiva in this population.

Several poor oral hygiene behaviours were associated with increased risk of preterm birth, including extended teeth cleaning time (≥30 min vs <5 min: RR 2.48, 95% CI: 1.41 to 4.39) and use of a non-traditional dentifrice (ie, sand, ash, oil or gul) (RR 2.22, 95% CI: 1.44 to 3.42). Teeth cleaning episodes per week (≥14 vs <7 times: RR 0.55, 95% CI: 0.29 to 1.02) was associated with decreased risk of

preterm birth. Risk of preterm birth increased with a dose response pattern for women who practiced multiple poor oral hygiene behaviours, defined as extended cleaning time per episode (≥5 min), infrequent cleaning episodes (<7 times per week) and use of a traditional dentifrice. Incidence of preterm birth was 17/208 (8.2%) for women with no poor behaviours, 155/1098 (14.1%) for women with one and 25/88 (28.4%) for those with two.

### Primary analysis: association between maternal gingival inflammation and preterm birth

Incidence of preterm birth by periodontal status was 113/840 (13.5%) for health and 84/554 (15.2%) for gingival inflammation. Incidence of preterm birth increased from 120/890 (13.5%) for women with either health (no sites BOP) or BOP <10% to 59/395 (14.9%) for women with localised gingival inflammation (BOP ≥10% and <30%) to 18/109 (16.5%) for women with generalised gingival inflammation (BOP ≥30%).

Our binary definition of gingival inflammation (BOP ≥10%) was not associated with preterm birth in the crude model (RR 1.13, 95% CI: 0.87 to 1.46) or after adjusting for confounders (aRR 1.16, 95% CI: 0.90 to 1.51). In the final adjusted model, increasing extent of gingival inflammation was associated with a non-significant increase in risk of preterm birth (BOP ≥30% vs no BOP: aRR 1.37, 95% CI: 0.81 to 2.32) (table 3). In this model, several oral hygiene behaviours were associated with preterm birth. Extended cleaning time (≥30 min vs <5 min: aRR

**Table 1** Participant characteristics at enrolment (n=1394)

| Characteristic | All | Term | Preterm | Mean difference/RR (95% CI)* |
|---|---|---|---|---|
| **Age** | | | | |
| Year (mean±SD) | 23.0±4.6 | 23.0±4.6 | 23.0±5.1 | 0.02 (−0.68 to 0.72) |
| **Age (years)** | | | | |
| <18 | 151 (10.8) | 124 (10.4) | 27 (13.7) | 1.33 (0.92 to 1.93) |
| 18 to <35 | 1216 (87.2) | 1053 (88.0) | 163 (82.7) | Ref |
| ≥35 | 27 (1.9) | 20 (1.7) | 7 (3.6) | **1.93 (1.01 to 3.72)** |
| **Ethnic group** | | | | |
| Hills (Pahadi) | 101 (7.3) | 89 (7.4) | 12 (6.1) | Ref |
| Plains (Madeshi) | 1292 (92.7) | 1107 (92.6) | 185 (93.9) | 1.21 (0.70 to 2.08) |
| **Height** | | | | |
| Cm (mean±SD) | 150.8±5.5 | 150.9±5.6 | 150.1±5.1 | −0.80 (−1.63 to 0.03) |
| **Weight** | | | | |
| Kg (mean±SD) | 45.8±7.0 | 46.0±7.1 | 44.3±5.8 | **−1.67 (−2.72 to −0.62)** |
| **BMI** | | | | |
| Underweight (<18.5 kg) | 418 (30.0) | 353 (29.5) | 65 (33.0) | 1.09 (0.83 to 1.44) |
| Normal weight (18.5 to <25 kg) | 893 (64.1) | 766 (64.0) | 127 (64.5) | Ref |
| Overweight or obese (≥25 kg) | 83 (6.0) | 78 (6.5) | 5 (2.5) | 0.42 (0.18 to 1.01) |
| **High blood pressure** | | | | |
| No | 1384 (99.3) | 1188 (99.2) | 196 (99.5) | Ref |
| Yes | 10 (0.7) | 9 (0.8) | 1 (0.5) | 0.71 (0.11 to 4.56) |
| **Gravidity** | | | | |
| First pregnancy | 395 (28.3) | 332 (27.7) | 63 (32.0) | 1.23 (0.92 to 1.63) |
| 1 to 3 previous pregnancies | 838 (60.1) | 729 (60.9) | 109 (55.3) | Ref |
| ≥4 pregnancies | 161 (11.5) | 136 (11.4) | 25 (12.7) | 1.19 (0.80 to 1.78) |
| **Urinary or vaginal infection†** | | | | |
| No | 1097 (78.7) | 942 (78.7) | 155 (78.7) | Ref |
| Yes | 297 (21.3) | 255 (21.3) | 42 (21.3) | 1.00 (0.73 to 1.37) |
| **Literacy** | | | | |
| No | 751 (53.9) | 638 (53.3) | 113 (57.4) | Ref |
| Yes | 643 (46.1) | 559 (46.7) | 84 (42.6) | 0.87 (0.67 to 1.13) |
| **Education (years)** | | | | |
| 0 | 751 (53.9) | 634 (53.0) | 117 (59.4) | Ref |
| 1–9 | 386 (27.7) | 343 (28.7) | 43 (21.8) | **0.72 (0.52 to 0.99)** |
| ≥10 | 256 (18.4) | 219 (18.3) | 37 (18.8) | 0.93 (0.66 to 1.31) |
| **Electricity** | | | | |
| No | 114 (8.2) | 97 (8.1) | 17 (8.6) | Ref |
| Yes | 1279 (91.8) | 1099 (91.9) | 180 (91.4) | 0.94 (0.60 to 1.49) |
| **House construction material** | | | | |
| None, thatch, sticks or bamboo | 858 (61.6) | 749 (62.6) | 109 (55.3) | Ref |
| Wood planks, bricks or stone | 535 (38.4) | 447 (37.4) | 88 (44.7) | **1.29 (1.00 to 1.68)** |
| **House roof material** | | | | |
| None, plastic, thatch or grass | 110 (7.9) | 92 (7.7) | 18 (9.1) | Ref |
| Tile, tin or concrete | 1283 (92.1) | 1104 (92.3) | 179 (90.9) | 0.85 (0.55 to 1.33) |
| **Latrine** | | | | |
| No latrine | 601 (43.1) | 515 (43.1) | 86 (43.7) | Ref |
| Brick, concrete or pit latrine | 792 (56.9) | 681 (56.9) | 111 (56.3) | 0.98 (0.75 to 1.27) |

Data presented as No. (%) unless otherwise noted.
Statistical significance assessed at p=0.05 level are in bold.
*T-test or relative risk and 95% CI as appropriate.
†Self-reported symptoms of painful urination or foul smelling vaginal discharge during pregnancy.
BMI, body mass index; RR, relative risk.

**Table 2** Periodontal disease status (n=1394)

| Characteristic | All | Term | Preterm | Rate of preterm % |
|---|---|---|---|---|
| **Health and gingival inflammation** | | | | |
| Health (all sites PD ≤3 mm and BOP <10%) | 840 (60.3) | 727 (60.7) | 113 (57.4) | 13.50 |
| Gingival inflammation (BOP ≥10% and/or PD ≥4 mm) | 554 (39.7) | 470 (39.3) | 84 (42.6) | 15.20 |
| Localised gingival inflammation (BOP <30%) | 445 (80.3) | 379 (80.6) | 66 (78.6) | 14.80 |
| Proportion with no sites PD ≥4 mm | 354 (79.6) | 299 (78.9) | 55 (83.3) | 15.50 |
| Proportion with ≥1 sites PD ≥4 mm | 91 (20.5) | 80 (21.2) | 11 (16.7) | 12.10 |
| Generalised gingival inflammation (BOP ≥30%) | 109 (19.7) | 91 (19.4) | 18 (21.4) | 16.50 |
| Proportion with no sites PD ≥4 mm | 80 (73.4) | 66 (72.5) | 14 (77.8) | 17.50 |
| Proportion with ≥1 sites PD ≥4 mm | 29 (26.6) | 25 (27.5) | 4 (22.2) | 13.80 |
| **Bleeding on probing (BOP)** | | | | |
| Per cent of sites BOP (mean±SD) | 10.2±12.3 | 10.1±12.4 | 10.6±12.0 | 14.1* |
| No sites BOP | 289 (20.7) | 249 (20.8) | 40 (20.3) | 13.80 |
| ≥1 site BOP | 1105 (79.3) | 949 (79.2) | 157 (79.7) | 14.20 |
| ≥1 site BOP and BOP<10% | 601 (43.1) | 521 (43.5) | 80 (40.6) | 13.30 |
| BOP ≥10% and BOP <30% | 395 (28.3) | 336 (28.1) | 59 (30.0) | 14.90 |
| BOP ≥30% | 109 (7.8) | 91 (7.6) | 18 (9.1) | 16.50 |
| **Probing depth (PD)** | | | | |
| Mean PD (mm) (mean±SD) | 1.7±0.3 | 1.7±0.3 | 1.7±0.3 | 14.1* |
| Mean PD at direct sites (mm) (mean±SD) | 1.5±0.3 | 1.5±0.3 | 1.5±0.2 | 14.1* |
| Per cent of sites PD ≥4 mm (mean±SD) | 0.2±1.0 | 0.2±1.0 | 0.1±0.9 | 14.1* |
| ≥1 site PD ≥4 mm | 120 (8.6) | 105 (8.8) | 15 (7.6) | 12.50 |
| **Clinical attachment loss (CAL)** | | | | |
| Mean CAL at direct sites (mm) (mean±SD) | 1.7±0.3 | 1.7±0.3 | 1.7±0.3 | 14.1* |
| ≥1 site recession ≥1 mm | 173 (12.4) | 154 (12.9) | 19 (9.6) | 11.00 |
| ≥1 site CAL ≥4 mm | 196 (14.1) | 172 (14.4) | 24 (12.2) | 12.20 |
| Per cent of CAL ≥4 mm due to pocketing (mean±SD) | 99.6±1.5 | 99.6±1.5 | 99.8±0.9 | 14.1* |
| Per cent of CAL ≥4 mm due to recession (mean±SD) | 0.4±1.5 | 0.4±1.5 | 0.2±0.9 | 14.1* |

Data presented as No. (%) unless otherwise noted.
*Preterm birth rate among all participants.
BOP, bleeding on probing.

2.44, 95% CI: 1.34 to 4.43), use of a traditional dentifrice (n=53, 3.9%) (aRR 2.17, 95% CI: 1.39 to 3.38) and fewer teeth cleaning episodes per week (≥14 times vs 1 to 6 times: aRR 0.46, 95% CI: 0.25 to 0.84) were associated with increased risk of preterm birth.

### Secondary analysis: stratification by gestational age at periodontal examination

In a secondary analysis, participants were stratified by their trimester at the time of the periodontal examination (first trimester: n=574; second trimester: n=820). Among women with dental exams in the first trimester, incidence of preterm birth increased from 47/381 (12.3%) for women with either health (no sites BOP) or BOP <10% to 28/158 (17.7%) for women with localised gingival inflammation (BOP ≥10% and <30%) to 8/35 (22.9%) for women with generalised gingival inflammation (BOP ≥30%). Our binary definition of gingival inflammation (BOP ≥10%) in the first trimester was associated with preterm (aRR 1.62, 95% CI: 1.08 to 2.42), but not in the second trimester (aRR 0.92, 95% CI: 0.65 to 1.30), after adjusting for confounders. In the final adjusted model stratified by trimester, there was a positive relationship between gingival inflammation and risk of preterm birth among women in the first trimester (BOP ≥30% vs no BOP: aRR 2.57, 95% CI: 1.11 to 5.95), but not among women in their second trimester (BOP ≥30% vs no BOP: aRR 1.05, 95% CI: 0.52 to 2.11) (table 4).

Our secondary exposure definition for mild periodontitis was not associated with preterm in a binary comparison or the crude or adjusted regression models for our primary or secondary analyses. These analyses were not conducted for moderate periodontitis given the low prevalence of the condition in this study population.

### DISCUSSION

In our community-based, prospective cohort study, gingival inflammation was an independent risk factor

**Table 3** Adjusted relative risks (aRR) between maternal gingival inflammation and preterm birth

| Characteristic | Unadjusted model (n=1394) | Preterm birth aRR (95% CI) | | |
| --- | --- | --- | --- | --- |
| | | Model 1 (n=1393) | Model 2 (n=1358) | Final model (n=1357) |
| **Gingival inflammation** | | | | |
| No sites BOP | Ref | Ref | Ref | Ref |
| ≥1 site BOP and BOP <10% | 0.96 (0.68 to 1.37) | 0.99 (0.70 to 1.40) | 1.08 (0.76 to 1.54) | 1.07 (0.75 to 1.51) |
| BOP ≥10% and BOP <30% | 1.08 (0.74 to 1.57) | 1.11 (0.77 to 1.61) | 1.19 (0.82 to 1.73) | 1.17 (0.80 to 1.69) |
| BOP ≥30% | 1.19 (0.72 to 1.99) | 1.24 (0.74 to 2.08) | 1.34 (0.79 to 2.28) | 1.37 (0.81 to 2.32) |
| **Age (years)** | | | | |
| <18 | | 1.21 (0.80 to 1.85) | 1.17 (0.77 to 1.76) | 1.21 (0.80 to 1.84) |
| 18 to <35 | | Ref | Ref | Ref |
| ≥35 | | **2.17 (1.14 to 4.11)** | 1.82 (0.95 to 3.49) | 1.65 (0.88 to 3.10) |
| **Pahadi/Madeshi** | | | | |
| Pahadi | | Ref | Ref | Ref |
| Madeshi | | 1.13 (0.65 to 1.97) | 0.95 (0.55 to 1.66) | 0.97 (0.55 to 1.70) |
| **BMI (kg)** | | | | |
| Underweight (<18.5) | | 1.10 (0.83 to 1.45) | 1.11 (0.85 to 1.47) | 1.15 (0.87 to 1.52) |
| Normal weight (18.5 to <25) | | Ref | Ref | Ref |
| Overweight or obese (≥25) | | 0.45 (0.18 to 1.08) | 0.49 (0.20 to 1.18) | 0.47 (0.20 to 1.13) |
| **Primiparous** | | | | |
| No | | Ref | Ref | Ref |
| Yes | | 1.17 (0.86 to 1.60) | 1.25 (0.92 to 1.71) | 1.26 (0.91 to 1.75) |
| **Multiple births** | | | | |
| No | | Ref | Ref | Ref |
| Yes | | **4.46 (2.88 to 6.92)** | **3.76 (2.46 to 5.75)** | **3.38 (2.09 to 5.46)** |
| **Teeth cleaning time per episode (min)** | | | | |
| <5 | | | Ref | Ref |
| 5 to <30 | | | **1.77 (1.07 to 2.94)** | **1.83 (1.11 to 3.04)** |
| ≥30 | | | **2.37 (1.31 to 4.27)** | **2.44 (1.34 to 4.43)** |
| **Teeth cleaning episodes per week** | | | | |
| 1 to 6 | | | Ref | Ref |
| 7 to 13 | | | 0.64 (0.39 to 1.05) | **0.62 (0.39 to 1.00)** |
| ≥14 | | | **0.48 (0.26 to 0.89)** | **0.46 (0.25 to 0.84)** |
| **Use of traditional dentifrice** | | | | |
| No | | | Ref | Ref |
| Yes | | | **2.13 (1.37 to 3.31)** | **2.17 (1.39 to 3.38)** |
| **Literacy** | | | | |
| No | | | | Ref |
| Yes | | | | 1.51 (0.85 to 2.66) |
| **Education (years)** | | | | |
| 0 | | | | Ref |
| 1 to 9 | | | | **0.50 (0.27 to 0.92)** |
| ≥10 | | | | 0.65 (0.35 to 1.21) |
| **House construction material** | | | | |
| None, plastic, thatch or grass | | | | Ref |
| Wood planks, bricks or stone | | | | **1.47 (1.11 to 1.93)** |
| **Roof construction material** | | | | |
| None, plastic, thatch or grass | | | | Ref |
| Tin, tile or concrete | | | | 0.85 (0.55 to 1.33) |

Statistical significance assessed at p=0.05 level are in bold.
BMI, body mass index; BOP, bleeding on probing.

**Table 4** Adjusted relative risks (aRR) between maternal gingival inflammation and preterm birth stratified by trimester at periodontal examination visit

| Characteristic | Preterm birth aRR (95% CI) | |
|---|---|---|
| | Women in first trimester (n=559) | Women in second trimester (n=798) |
| Gingival inflammation | | |
| No sites BOP | Ref | Ref |
| ≥1 site BOP and BOP <10% | 1.51 (0.81 to 2.84) | 0.93 (0.60 to 1.44) |
| BOP ≥10% and BOP <30% | **1.91 (1.00 to 3.63)** | 0.88 (0.55 to 1.42) |
| BOP ≥30% | **2.57 (1.11 to 5.95)** | 1.05 (0.52 to 2.11) |
| Age (years) | | |
| <18 | 1.06 (0.52 to 2.18) | 1.42 (0.85 to 2.36) |
| 18 to <35 | Ref | Ref |
| ≥35 | 0.97 (0.28 to 3.38) | **2.42 (1.14 to 5.14)** |
| Pahadi/Madeshi | | |
| Pahadi | Ref | Ref |
| Madeshi | 1.15 (0.46 to 2.90) | 0.82 (0.42 to 1.60) |
| BMI (kg) | | |
| Underweight (<18.5) | 1.44 (0.95 to 2.18) | 0.96 (0.65 to 1.42) |
| Normal weight (18.5 to <25) | Ref | Ref |
| Overweight or obese (≥25) | 0.72 (0.19 to 2.78) | 0.33 (0.11 to 1.06) |
| Primiparous | | |
| No | Ref | Ref |
| Yes | 1.41 (0.84 to 2.34) | 1.13 (0.74 to 1.72) |
| Multiple births | | |
| No | Ref | Ref |
| Yes | 2.57 (0.92 to 7.15) | **4.38 (2.57 to 7.47)** |
| Teeth cleaning time per episode (min) | | |
| <5 | Ref | Ref |
| 5 to <30 | 1.40 (0.71 to 2.75) | **2.42 (1.13 to 5.19)** |
| ≥30 | 2.06 (0.92 to 4.57) | **2.93 (1.19 to 7.23)** |
| Teeth cleaning episodes per week | | |
| 1 to 6 | Ref | Ref |
| 7 to 13 | 0.87 (0.39 to 1.93) | **0.49 (0.28 to 0.85)** |
| ≥14 | 0.49 (0.18 to 1.31) | **0.46 (0.22 to 0.95)** |
| Use of traditional dentifrice | | |
| No | Ref | Ref |
| Yes | 1.61 (0.63 to 4.10) | **2.58 (1.56 to 4.26)** |
| Literacy | | |
| No | Ref | Ref |
| Yes | 1.97 (0.78 to 4.96) | 1.25 (0.60 to 2.61) |
| Education (years) | | |
| 0 | Ref | Ref |
| 1–9 | **0.34 (0.13 to 0.89)** | 0.67 (0.31 to 1.45) |
| ≥10 | 0.41 (0.15 to 1.11) | 0.93 (0.41 to 2.10) |

Continued

**Table 4** Continued

| Characteristic | Preterm birth aRR (95% CI) | |
|---|---|---|
| | Women in first trimester (n=559) | Women in second trimester (n=798) |
| House construction material | | |
| None, plastic, thatch or grass | Ref | Ref |
| Wood planks, bricks or stone | 1.33 (0.84 to 2.11) | **1.57 (1.09 to 2.25)** |
| Roof construction material | | |
| None, plastic, thatch or grass | Ref | Ref |
| Tin, tile or concrete | 0.80 (0.39 to 1.65) | 0.90 (0.51 to 1.59) |

Statistical significance assessed at p=0.05 level are in bold.
BMI, body mass index; BOP, bleeding on probing.

for preterm birth among women examined during their first, but not second, trimester, after adjusting for potential confounding risk factors. Our finding suggests that gingival infection and inflammatory burden early in pregnancy, potentially originating prior to conception, could be responsible for the observed relationship, which becomes obscured in the second trimester by the presence of women with initially healthy periodontal conditions who develop pregnancy-induced gingival inflammation. This result is consistent with studies that have reported that active, and especially progressive, periodontal infection is most harmful early in pregnancy.[13] This also supports the hypothesis, proposed to explain the mixed results of RCTs of this association, that intervening on women with periodontal disease later in pregnancy, or even after conception, may be too late to affect the proposed causal pathway.[33]

Most previous studies have considered the relationship between varying definitions and severities of periodontitis and preterm birth. A meta-analysis of seven cohort studies estimated a pooled relative risk of preterm of 1.70 (95% CI: 1.03 to 2.81) for pregnant women with periodontitis versus health.[10] Prevalence of periodontitis in our study population was low, although not dissimilar to previous surveys of periodontal conditions among this age group in Nepal.[34] The low prevalence of periodontitis in this population is likely attributable to the population's low age (median 22.2, IQR: 19.7 to 25.5; mean 23.0, SD: 4.6) and absence of other common risk factors for the condition. An unpublished review of community-based studies of the periodontal disease and preterm birth relationship in low- and middle-income countries conducted by our research team found mean maternal ages ranging from 26 to 29. A previous subanalysis, which was nested within this cohort and conducted by our research team, assessed various potential risk factors for gingivitis in pregnant women in this population, identifying maternal age, maternal short stature, Pahadi ethnicity (vs Madeshi) and some indicators of low socioeconomic status and poor oral

hygiene behaviours as significant.[32] Unlike the majority of studies of this association, particularly in high-income countries, the prevalence of important confounders, such as smoking, alcohol use and morbidities associated with chronic disease, were nearly absent in our study population. An association between periodontitis and preterm birth may have been seen in our population had there been a higher prevalence and severity of periodontitis, as would be expected in an older population.

Few observational or interventional studies have considered gingival inflammation as the primary exposure definition when investigating the relationship between periodontal disease and preterm birth. A cohort study by Kruse et al[35] found an association between gingivitis and high risk of preterm birth among women without periodontitis in a hospital setting in Germany. A trial by López et al[36] of women with gingivitis in a hospital setting in Santiago, Chile, demonstrated a significantly higher risk of preterm low birth weight among women with gingivitis who received periodontal treatment after delivery compared with those that received treatment during pregnancy (<28 weeks) (aOR: 2.76, 95% CI: 1.29 to 5.88). Gingival inflammation requires less intensive treatment than periodontitis, and can include improved oral hygiene, use of an antiseptic oral rinse and periodontal therapy, interventions that could be feasibly delivered in a rural community setting. While antiseptic oral rinses are known for their ability to reduce plaque and control inflammation, their action on pregnancy outcomes is underexplored.[37 38] However, Jeffcoat et al[39] found a reduction in preterm birth in a high-risk population with the use of cetylpyridinium chloride oral rinse intervention.

Extent and severity of gingival inflammation, independent of traditional clinical measures of periodontitis, such as CAL or PD, have been associated with magnitude of bacteraemia.[40] Transient bacteraemia, facilitated by ulceration, inflammation and the increased vascular permeability of the gingiva, could pose increased risk for adverse pregnancy outcomes.[41] In this case, treatment of traditional clinical measures of periodontitis—without reducing infection, inflammation and, potentially, the presence of specific harmful pathogens—could fail to disrupt the proposed causal pathway between exposure and outcome. Further, if such bacteraemia were to occur early in pregnancy, treatment of clinical conditions later in pregnancy would be too late to eliminate exposure of the placenta and/or fetus to these pathogens, and hence would not decrease of risk of adverse pregnancy outcomes.[41]

Poor oral hygiene behaviours, including extended teeth cleaning time (≥5 min), infrequent teeth cleaning (<7 times per week) and use of traditional dentifrice (ie, sand, ash, oil or gul) or datiwan, were also associated with increased risk of preterm birth in our analysis. Studies have shown that the mechanical manipulation involved in brushing, particularly frequent or forceful brushing or use of brushes with hard filaments, can lead to gingival

abrasion, recession and bacteraemia.[42–44] Some have posited that bacteraemia caused by mechanical manipulation of the gingiva involved in periodontal therapy may increase risk of adverse pregnancy outcomes, attenuating observation of any true effect in interventional studies of this association.[33] More research is needed to understand the extent to which poor oral hygiene behaviours in pregnant women, including common traditional practices, such as datiwan use, could—via local inflammatory responses or increased access of periodontal bacteria or their byproducts to the bloodstream—initiate a cascade of effects that ultimately influences the pregnancy outcome.

An important strength of our study was that we used a population-based sample, compared with previous observational studies of this relationship, which were primarily conducted in health facility settings, introducing risk of selection bias associated with the likelihood of women in the target population delivering in a particular facility. The potential for selection bias to influence the measure of association in a facility-based study is especially high in populations where a substantial proportion of women deliver at home, as is the case in this area of rural Nepal, where roughly half of women deliver in a facility. In the unpublished review conducted by our research team, we found one study of this association, by Mobeen et al,[45] that used a population-based sample. Conducted in a peri-urban area of Hyderabad, Pakistan, the authors reported significant associations of varying degrees between measures of periodontal disease and neonatal death, perinatal death and stillbirth.[45]

A limitation of this study was the collection of clinical recession measures from only the direct buccal and lingual surfaces. In the absence of these data, we likely underestimated the burden of periodontal disease among pregnant women in this study, potentially attenuating our measure of association. We were unable to control for some confounders of this relationship, including previous preterm birth, certain chronic diseases (eg, diabetes), and we only used a proxy (self-reported symptoms) for urinary tract and vaginal infections. Conducting periodontal exams in participant homes, where light and other conditions are variable, may have introduced a level of measurement error in our exposure that could not be quantified or controlled for given the setting and logistical constraints of our study. Lastly, preterm birth was based on maternal self-report of LMP instead of ultrasound examination. If our LMP estimates were less accurate among women examined later in pregnancy, due to longer maternal recall relative to women examined earlier in pregnancy, this could have resulted in non-differential misclassification of our binary outcome, preterm birth, which would tend to attenuate observation of a true relationship among women examined in the second trimester.

## CONCLUSION

Our study identified gingival inflammation as an independent risk factor for preterm birth among women

examined early in pregnancy. Women non-compliant with oral hygiene care recommendations were at substantially increased risk of preterm birth. These findings highlight the importance of adherence to proper home oral self-care in pregnant women and women expecting to become pregnant. Oral health policies and education programmes should include tailored approaches to encourage improved self-care practices in this important population. Future studies should evaluate the effectiveness of community-based oral health interventions that specifically target gingival inflammation—delivered to women early in or prior to pregnancy—on the incidence of preterm birth and other adverse pregnancy outcomes in low-income countries with high risk for these outcomes.

**Acknowledgements** The authors thank the auxiliary nurse midwives who conducted the periodontal examinations: Kaushila Chaudhary, Lalita Lama, Gayatri Mahat, Shanti Sharma and Santoshi Kumari. Thank you to Roshan Chaudhary for serving as the laboratory technician. We also wish a special thank you to the women from Sarlahi District, Nepal, for participating in this study. This study was carried out in close collaboration with our implementing partner, Nepal Netra Jyoti Sangh, under the auspices of the Social Welfare Council of the Government of Nepal.

**Contributors** DJE conceptualised and designed the study, developed field implementation protocols, led data collection in the field, conducted the analysis and wrote the manuscript. SKK conceptualised and designed the study, oversaw field implementation and ensured quality data collection and provided comments on the manuscript. NKA conceptualised and designed the study, trained and oversaw the data collectors and provided comments on the manuscript. JK conceptualised and designed the study, ensured quality data collection, advised on analytical approach and provided comments on the manuscript. SCL conceptualised and designed the study, supported overall implementation in the field and provided comments on the manuscript. BR contributed to field implementation, ensured quality control of field procedures and provided comments on the manuscript. MAR conceptualised and designed the study, contributed to the analysis and interpretation of the results and provided comments on the manuscript. LCM conceptualised and designed the study, obtained funding for the study, oversaw implementation of data collection, obtained ethical approvals, advised on analytical approach and provided edits and comments on the manuscript. All authors approved the final manuscript as submitted and agree to be accountable for all aspects of the work.

**Funding** This work was supported by the National Institute for Child Health and Development (HD060712) and the Bill & Melinda Gates Foundation (OPP1084399, OPP1131701).

**Competing interests** None declared.

**Patient consent for publication** Not required.

**Ethics approval** The study received ethical approval from the Institutional Review Board at Johns Hopkins Bloomberg School of Public Health (Baltimore, USA) and the Ethical Review Board of the Nepal Health Research Council (Kathmandu, Nepal).

**Provenance and peer review** Not commissioned; externally peer reviewed.

**Data availability statement** Data are available in a public, open access repository. All data files, codebooks and related manuscripts are available from the JHU Data Archive (https://doi.org/10.7281/T1/ZPGBJW).

**ORCID iD**
Daniel J Erchick http://orcid.org/0000-0002-2852-280X

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
