## [Reviewer comments · BMJ Open]

ARTICLE DETAILS

TITLE (PROVISIONAL)	Risk of preterm birth associated with maternal gingival inflammation and oral hygiene behaviors in rural Nepal: A community-based, prospective cohort study
AUTHORS	Erchick, Daniel; Khatry, Subarna; Agrawal, Nitin; Katz, Joanne; LeClerq, Steven; Rai, Bhola; Reynolds, Mark; Mullany, Luke C.

VERSION 1 – REVIEW

REVIEWER	Erin Hartnett NYU Rory Meyers College of Nursing USA
REVIEW RETURNED	08-Jan-2020

GENERAL COMMENTS	No mention of consent for this study In abstract - Describe objectives and method more clearly
---

REVIEWER	Adela Sembaj Biochemistry and Molecular department, School of Medicine, National University of Cordoba, Argentina
REVIEW RETURNED	17-Feb-2020

GENERAL COMMENTS	terai, will be with the first capital letter, Terai. It is important to note that the periodontal state and low birth weight and/or preterm birth have common risk factors in their causal chain, which can act synergistically or as modifiers of this association. From this point of view, PD can be considered a risk factor for pre-term pregnancies and / or born with low weight. Solid studies on this subject, with strict control of confounding factors, showed different results. This article shows the results in a specific area. The study was very well designed y take into account all the cofounders' factors. The introduction presents the background grounded. The objective is clearly presented. The experimental design is described; the places of recruitment are described. The variables to be analyzed and the inconveniences that arose in certain situations are stated. Statistical methods are stated. They use a flow chart to describe the number of patients included in this study The conclusion of the study is that gingival inflammation is an independent risk factor for preterm birth among women examined early in pregnancy. For this reason, it would need to promote public policy to improve the oral health of women before pregnancy in special in poor countries. This article is well designed but the results are the expected.
--

REVIEWER	Ina Schüler
-----------------	-------------

	Jena University Hospital, Germany
REVIEW RETURNED	25-Mar-2020
GENERAL COMMENTS	I would like to congratulate the authors for this manuscript, which is addressing a relevant issue and clearly written. Limitations are critically discussed. Thanks for your work.
REVIEWER	Prof CWJ Africa University of the Western Cape South Africa
REVIEW RETURNED	28-Mar-2020
GENERAL COMMENTS	This study involved a lot of work for which I commend the authors. However, it is not clear to me exactly how it differs from the earlier publication cited as 31 in the reference list. It appears to be a re-arrangement of the contents of the earlier study. In addition, the grammar needs attention, particularly related to inserting "a" and "the" Some sentences are too long with the result that the reader has to re-read (in some cases what is essentially a full paragraph) to grasp its meaning. page 8, line 8 should read 27-29 and not 27,29 (what happened to 28?) page 10, first paragraph belongs in the methodology. The methodology is not clearly explained (except for the actual clinical measurements which are clear). The entire paper lacks proper structure and separation of relevant sections. This should be addressed.

VERSION 1 – AUTHOR RESPONSE

Reviewer: 1
Reviewer Name
Erin Hartnett
Institution and Country
NYU Rory Meyers College of Nursing USA

Please state any competing interests or state 'None declared': None declared

Please leave your comments for the authors below

1. No mention of consent for this study

Thank you for your comments. As mentioned above in response to the editorial request, we have added a sentence to the last paragraph of the methods section indicating that written consent was obtained from all participants in this study.

2. In abstract - Describe objectives and method more clearly

We have revised methods section of the abstract to more clearly articulate our primary research aim.

Reviewer: 2

Reviewer Name

Adela Sembaj

Institution and Country

Biochemistry and Molecular department, School of Medicine, National University of Cordoba, Argentina

Please state any competing interests or state 'None declared': None declared

Please leave your comments for the authors below

1. terai, will be with the first capital letter, Terai.

Thank you for your comments. We have addressed this error in the text.

2. It is important to note that the periodontal state and low birth weight and/or preterm birth have common risk factors in their causal chain, which can act synergistically or as modifiers of this association. From this point of view, PD can be considered a risk factor for pre-term pregnancies and / or born with low weight. Solid studies on this subject, with strict control of confounding factors, showed different results. This article shows the results in a specific area.

We have addressed this point in the introduction by elaborating further on the previously described risk factors for the association between periodontal disease and adverse pregnancy outcomes. Please see changes made to the fourth paragraph in the Introduction Section.

3. The study was very well designed y take into account all the cofounders' factors. The introduction presents the background grounded. The objective is clearly presented. The experimental design is described; the places of recruitment are described. The variables to be analyzed and the inconveniences that arose in certain situations are stated. Statistical methods are stated. They use a flow chart to describe the number of patients included in this study. The conclusion of the study is that gingival inflammation is an independent risk factor for preterm birth among women examined early in pregnancy. For this reason, it would need to promote public policy to improve the oral health of women before pregnancy in special in poor countries. This article is well designed but the results are the expected.

We have included a sentence in the Conclusion Section after the Discussion to recognize the need to encourage policies and education programs to improve oral self-care in this population.

Reviewer: 3

Reviewer Name

Ina Schüler

Institution and Country

Jena University Hospital, Germany

Please state any competing interests or state 'None declared': None declared

Please leave your comments for the authors below

1. I would like to congratulate the authors for this manuscript, which is addressing a relevant issue and clearly written. Limitations are critically discussed. Thanks for your work.

Thank you for your positive comments.

Reviewer: 4
Reviewer Name
Prof CWJ Africa
Institution and Country
University of the Western Cape South Africa

Please state any competing interests or state 'None declared': None declared

Please leave your comments for the authors below

1. This study involved a lot of work for which I commend the authors. However, it is not clear to me exactly how it differs from the earlier publication cited as 31 in the reference list. It appears to be a re-arrangement of the contents of the earlier study.

Thank you for your helpful comments. The earlier study was a cross-sectional survey with the aim of describing the oral health status and oral health behaviors of the population. The aim of this prospective cohort study was to estimate associations between gingivitis and preterm birth in this population. We have made edits to the Abstract and Methods Sections to clarify this point.

2. In addition, the grammar needs attention, particularly related to inserting "a" and "the". Some sentences are too long with the result that the reader has to re-read (in some cases what is essentially a full paragraph) to grasp its meaning.
page 8, line 8 should read 27-29 and not 27,29 (what happened to 28?)
page 10, first paragraph belongs in the methodology.

Thank you for these points. We have done a copy edit of the manuscript to improve clarity. With regard to your specific point on page 8, line 8: Reference 28 was previously cited above (page 7, line 52). The sentence on page 8, line 8, only requires citation of references 27 and 29; hence we have not included reference 28. In regard to your point on page 10, first paragraph: We think describing the actual numbers of participants enrolled, lost to follow-up, etc. is best done in the beginning of the results section. In the methods section, we describe our sample size calculation and rationale.

3. The methodology is not clearly explained (except for the actual clinical measurements which are clear). The entire paper lacks proper structure and separation of relevant sections. This should be addressed.

We have tried to improve the flow and clarity through a full edit of the manuscript.

VERSION 2 – REVIEW

REVIEWER	Adela Sembaj Biochemical and Molecular Biology Department, School of Medicine, National University of Cordoba, Argentina
REVIEW RETURNED	20-May-2020
GENERAL COMMENTS	I congratulate the authors for the work, it has required a lot of effort. The community where they work deserves this type of effort. For the entire population, they should promote improved dental health through dental hygiene campaigns.